# Deep Learning Chest CT for Clinically Precise Prediction of Sepsis-Induced Acute Respiratory Distress Syndrome: A Protocol for an Observational Ambispective Cohort Study

**DOI:** 10.3390/healthcare10112150

**Published:** 2022-10-28

**Authors:** Han Li, Yang Gu, Xun Liu, Xiaoling Yi, Ziying Li, Yunfang Yu, Tao Yu, Li Li

**Affiliations:** 1The Second Clinical Medical College, Southern Medical University, Guangzhou 510515, China; 2Department of Emergency Medicine, Sun Yat-sen Memorial Hospital, Sun Yat-sen University, Guangzhou 510289, China; 3Department of Anesthesiology, Shenshan Medical Center, Memorial Hospital of Sun Yat-sen University, Shanwei 516621, China; 4Department of Intensive Care Medicine, Sun Yat-sen Memorial Hospital, Sun Yat-sen University, Guangzhou 510289, China; 5Department of Medical Oncology, Sun Yat-sen Memorial Hospital, Sun Yat-sen University, Guangzhou 510289, China; 6Faculty of Medicine, Macau University of Science and Technology, Taipa, Macao, China

**Keywords:** radiomics, sepsis, acute respiratory distress syndrome, prediction model

## Abstract

**Background:** Sepsis commonly causes acute respiratory distress syndrome (ARDS), and ARDS contributes to poor prognosis in sepsis patients. Early prediction of ARDS for sepsis patients remains a clinical challenge. This study aims to develop and validate chest computed tomography (CT) radiomic-based signatures for early prediction of ARDS and assessment of individual severity in sepsis patients. **Methods:** In this ambispective observational cohort study, a deep learning model, a sepsis-induced acute respiratory distress syndrome (SI-ARDS) prediction neural network, will be developed to extract radiomics features of chest CT from sepsis patients. The datasets will be collected from these retrospective and prospective cohorts, including 400 patients diagnosed with sepsis-3 definition during a period from 1 May 2015 to 30 May 2022. 160 patients of the retrospective cohort will be selected as a discovering group to reconstruct the model and 40 patients of the retrospective cohort will be selected as a testing group for internal validation. Additionally, 200 patients of the prospective cohort from two hospitals will be selected as a validating group for external validation. Data pertaining to chest CT, clinical information, immune-associated inflammatory indicators and follow-up will be collected. The primary outcome is to develop and validate the model, predicting in-hospital incidence of SI-ARDS. Finally, model performance will be evaluated using the area under the curve (AUC) of receiver operating characteristic (ROC), sensitivity and specificity, using internal and external validations. **Discussion:** Present studies reveal that early identification and classification of the SI-ARDS is essential to improve prognosis and disease management. Chest CT has been sought as a useful diagnostic tool to identify ARDS. However, when characteristic imaging findings were clearly presented, delays in diagnosis and treatment were impossible to avoid. In this ambispective cohort study, we hope to develop a novel model incorporating radiomic signatures and clinical signatures to provide an easy-to-use and individualized prediction of SI-ARDS occurrence and severe degree in patients at early stage.

## 1. Introduction

Sepsis-induced acute respiratory distress syndrome (SI-ARDS) is a common and severe complication of sepsis and is an independent contributor to poor prognosis of patients [1,2]. Acute respiratory distress syndrome (ARDS) is characterized by lung epithelial injury with a decrease of oxygenation index and is manifested as respiratory distress [3]. Once injury to the lung epithelial damage is evidently promoted, the progression to ARDS cannot be reversed, and the mortality rate reaches approximately 40% [4,5]. However, it remains a clinical challenge to identify SI-ARDS early and accurately, something which could optimize treatment strategy and reduce the mortality risk [6]. As such, a novel method for early identification of SI-ARDS is of paramount interest to clinicians and researchers.

Immune dysfunction driving sepsis-induced target organ damage has been fully appreciated and is characterized by excessive inflammatory reaction and immune function inhibition [7,8]. However, despite much significant research that has been undertaken to improve the understanding of ARDS pathogenesis, including viable predictive molecular biomarkers and multigene-based expression assays, early identification for SI-ARDS remains elusive [9,10,11,12,13]. Additionally, the immune microenvironment, influenced by the above mechanisms, has not been quantitatively assessed for the early identification of SI-ARDS, which comprises immune cell subsets, inflammatory cytokine and other components [14,15,16]. Recent studies have revealed artificial intelligence with radiomics emerging as a quantitative imaging method for early detection, risk assessment, and treatment decisions in various diseases [17,18]. Despite the lack of a clear correlation between radiomics and pathophysiology, application of radiomics in predicting SI-ARDS is worth exploring.

Computerized tomography (CT) of the chest is easily accessible for patients in order for the identification of pulmonary or non-pulmonary sepsis and is also a method for ARDS diagnosis based on characteristic radiologic features. Also, radiomic high-dimensional features extracted from CT images offer an insight into microvascular injury of SI-ARDS, which are imperceptible to human eyes and relevant to intra-alveolar heterogeneity and potential prognosis. Herein, we used a single center ambispecive cohort study to develop a radiomics classification model and demonstrated the robust correlation between radiologic features and SI-ARDS occurrence. Further, we established a prediction model integrating radiomics and clinicopathological characteristics for more precise prediction of ARDS severity.

## 2. Materials and Methods

### 2.1. Study Design

The study was registered with ClinicalTrials.gov (NCT04541264, https://clinicaltrials.gov/show/NCT04541264) on 9 September 2020. In this double-center, ambispective, observational study, we will collect chest CT and clinical information of patients with sepsis and septic shock treated in our hospital (Sun Yat-sen Memorial Hospital, Sun Yat-sen University) and Shenshan Medical Center, Memorial Hospital of Sun Yat-sen University (from 1 May 2015 to 30 May 2022). For the final analysis, 400 patients will be enrolled according to the TRIPOD guideline [19]. Among the 400 patients, 200 will be retrospectively recruited from our hospital from May 2015 to August 2020, 100 patients will be prospectively recruited from our hospital from August 2020 to May 2022, and the other 100 patients will be prospectively recruited from Shenshan Medical Center, Memorial Hospital of Sun Yat-sen University from August 2020 to May 2022. One hundred-sixty patients of the retrospective cohort will be selected as a discovering group to reconstruct the model, and 40 patients of the retrospective cohort will be selected as a testing group for internal validation. Additionally, 200 patients of the prospective cohort will be selected as a validating group for external validation. Follow-up of the relevant patients will be conducted until April 2023. This study was approved on 19 August 2020 by the Ethics Committee of Sun Yat-sen Memorial Hospital, Sun Yat-sen University, Guangzhou, China. Figure 1 shows the flow chart of the study protocol.

### 2.2. Cohort Descriptions and Definitions

This study will recruit 200 patients in a developing group and 200 patients in a validating group from two hospitals. Patients’ age above 18 years diagnosed as sepsis or septic shock will be screened for eligibility. SI-ARDS is defined by sequential occurrence of the sepsis-3 consensus criteria for sepsis [7] and the Berlin definition for ARDS [2]. Exclusion criteria are (1) admission stay <24 h, (2) patients with end-stage lung disease or long-term oxygen therapy, (3) patients with sepsis or septic shock whose oxygenation index (OI, arterial oxygen tension/inspired oxygen fraction [PaO2/FIO2]) ≤300 mmHg before admission, (4) a history of lung transplantation and chronic obstructive pulmonary disease, and (5) cancer patients who have/have not received chemotherapy. Since sepsis and ARDS are not easy to diagnose, two doctors will separately perform patient inclusion/exclusion criteria, extract the data, and assess the data quality. Any discrepancies will be resolved by consensus, and if necessary, a third professor can be consulted.

### 2.3. Sample Size Calculation

The sample size calculation based on logistic regression analysis approximately require 20 variables, regarded as risk factors for SI-ARDS, which are selected from radiomic features based on a previous study [20]. Meanwhile, the number of sepsis patients should be 5–10 times the number of risk factors. Considering a 25–50% occurrence rate of ARDS in sepsis patients [21,22], we calculate a minimum sample size of 160 patients for the discovering group.

### 2.4. Data Collection

All clinical data will be collected by investigators and trained personnel. Each participant’s data will be filled in electronic case report forms (CRF) and stored online using research electronic data capture (REDCap) [23]. The CRF is designed to collect the necessary information based on this study protocol. The detailed contents of CRF contain four main parts: (1) demographics and chronic disease history (including age, gender, smoking status, hypertension, diabetes mellitus, chronic lung disease and coronary heart disease), (2) laboratory tests and imaging examination (including routine blood, routine biochemistry, coagulation function, arterial blood gas analysis, blood culture, sputum culture, and ID of chest CT), (3) treatment process (including source of infection, anti-infection therapy, ventilator use or not, and type and dose of vasopressor therapy), (4) outcomes (intensive care unit (ICU) stay or not, length of ICU stay, total length of hospital stay, and in-hospital death).

### 2.5. Work-Flow of Radiomics Analysis

To feed the data for the model training, we will initially perform the data pre-processing for CT images and clinical features. Since CT-scanning of an individual utilizes different equipment and parameter settings, inhomogeneities in voxel spacing of the CT DICOM image is very common. To facilitate the data processing of the convolutional neural networks, we will count each grid of pixels with x × y × z pixel dimensions. Then, taking the median of each direction as the standard, we will use bilinear interpolation in the X and Y directions and nearest neighbor interpolation in the Z direction. Based on the theory that organs quantified by CT image require intensity values, the same intensity value represents the same change when scanning the same organ. We will select the lung window settings as width 1500 HU and level −500 HU and take 5–95% of the pixel values to make the intensity value more easily fall in the range for which the convolutional neural network is working. To further enhance the accuracy, we intend to manually outline the region of interest (ROI) to reduce the input of useless information. For the clinical information form, we will first deal with missing values and outliers with methods such as filling in 0, filling in the average value or eliminating them. We will then normalize all numerical variables for input into the neural network model. The workflow of data pre-processing can be seen in Figure 2.

We intend to employ modified ResNeSt [24], DenseNet [25] and EfficientNet [26] models to train a robust model. All networks will use 3D convolution, ReLU/LeakyReLU nonlinear activation function and batch normalization. For the entire CT image and the outlined region of interest, we will train two convolutional neural networks to extract features. For the ROI, we will randomly sample the voxels containing 80% of the ROI from the entire CT image to generate an image. We will perform five-fold cross-validation on the training set in model training phase, and use random rotation, translation and transposition data enhancement strategies to increase the data and reduce the imbalance between classes. Additionally, the network will adopt the cross-entropy loss as a loss function, Xavier as initialized parameter and stochastic average gradient descent as optimizer. In the last fully connected layer of the network, we will modify the corresponding structure by adding pre-processed clinical variables and other data, and finally predicted whether ARDS might occur.

### 2.6. Outcome Measures

In this study, the primary outcome measure will be occurrence of ARDS, which is defined as pulmonary or nonpulmonary sepsis that had progressed into ARDS.

Secondary outcome measures are as follows:

(1)Grade of pulmonary damage: patients will be classified as mild (200 <  PaO2/FiO2 ≤ 300 mmHg), moderate (100 < PaO2/FiO2 ≤ 200 mmHg), or severe (PaO2/FiO2  ≤  100 mmHg) at the moment of diagnosis of ARDS [2];(2)Ventilator-free days, VFD: defined as the number of days between successful weaning from mechanical ventilation and day 28 after ICU admission;(3)Respiratory organ failure-free days, RFFD: defined as the number of days between a day without evidence of respiratory organ failure [27];(4)28-day mortality: all-cause mortality within 28 days following enrolment.

### 2.7. Statistical Analysis

Continuous variables between two groups will be compared by T test or Wilcoxon rank sum test, whereas categorical variables will be compared by χ^2^ test. Univariate and multivariate logistic regression will predict occurrence of ARDS, using variables that selected by consensus of variable importance ranking by random forest [28] and least absolute shrinkage and selector operation (LASSO) regression techniques [29]. With the selected clinical and radiomics features, we will develop two versions of the ARDS prediction model, using logistic regression (LR) and support vector machine (SVM). Then a prognostic model will be built for prediction of individual and classification of severe level. ROC curves will determine the optimal cut-off point for models, and patients will be divided into high-risk and low-risk groups. VFD, RFFD and 28-day mortality will be screened indicators with LASSO-COX and RFE-SVM algorithm. Thereafter, these will be estimated using the Kaplan–Meier method. All these algorithm performances will be assessed using the AUC and *p*-values less than 0.05 will be considered statistically significant. After these main analyses, other subgroups will be carried out as following: septic shock vs. non-septic shock, and pulmonary sepsis vs. non-pulmonary sepsis. Statistical analysis will be performed using the R programming language, including its packages.

## 3. Discussion

Mild ARDS patients, defined with OI ≤ 300 mmHg but >200 mmHg, may not require invasive mechanical ventilation, and the condition is more readily reversible [30]. To date, early lung-protective ventilation strategies with lower tidal volume ventilation that protect the lungs from the progression of injury, have revealed reduced mortality in ARDS patients [31,32]. Although SI-ARDS patients with end-stage refractory respiratory failure can be efficiently treated by venovenous extracorporeal membrane oxygenation (VV-ECMO), surviving patients with severe ARDS may progress to pulmonary fibrosis, resulting in a progressive decline of respiratory function and ending in death [33]. Thus, an early identification and classification of the SI-ARDS is essential to improve patients’ prognosis and disease management.

Until now, OI and radiologic features have been sought as classical indices to identify ARDS in sepsis patients. However, when OI declines and characteristic imaging findings are clearly presented, delays in diagnosis and treatment may be impossible to avoid. Two previous studies have tried to utilize the lung injury prediction score (LIPS) to predict risk for ARDS by collecting routinely available clinical data, but this LIPS model has a relatively low positive predictive value for ARDS and depends on a reliable portable chest radiograph interpretation [6,34]. Other multigene-based expression assays are also currently used to predict early SI-ARDS but their predictive utilities are still restricted in certain patient subsets who are characterized with the “reactive” phenotype [35,36,37].

Recently, artificial intelligence-based radiomics applied in predicting patients who are at risk of developing immunotherapy-induced pneumonitis and radiation pneumonitis have shown promising results [20,38]. Based on these findings, future studies could focus on comparing predictive values of different imaging approaches, such as CT, ultrasound and bronchoscopy combined with artificial intelligence technology. In this ambispective cohort study, we developed and validated a novel model incorporating radiomic signatures and clinical signatures to provide an easy-to-use and individualized prediction of SI-ARDS occurrence and severe degree in patients at early stage. Moreover, research has indicated that immune microenvironment abnormalities play a critical role in SI-ARDS [39,40], revealing that disturbances of neutrophil migration and neutrophil apoptosis caused neutrophil accumulation in the lung tissues, resulting in lung injury. We considered and initially explored correlation between radiomic features and immune-associated inflammatory indicators.

This study has several limitations. First, there are intrinsic limitations and possible patient selection bias in this observational and partly retrospective cohort study. Since the majority of sepsis patients first receive a chest X-ray, those with a relatively high risk of ARDS would be referred for chest CT and those with severe septic shock would receive bedside chest radiograph. However, a subgroup analysis will be performed by comparing mild sepsis group alone (removing the septic shock subgroup) with the pulmonary and non-pulmonary sepsis group. Second, the follow-up duration is relatively short. However, this period is quite adequate for consideration of the early primary outcome of SI-ARDS. A third limitation is the relatively small sample size and that high-dimensional extracted radiomics features may result in model overfitting. Therefore, these available data still remain inconclusive and need to be further validated in future multicenter prospective studies.

## Figures and Tables

**Figure 1 healthcare-10-02150-f001:**
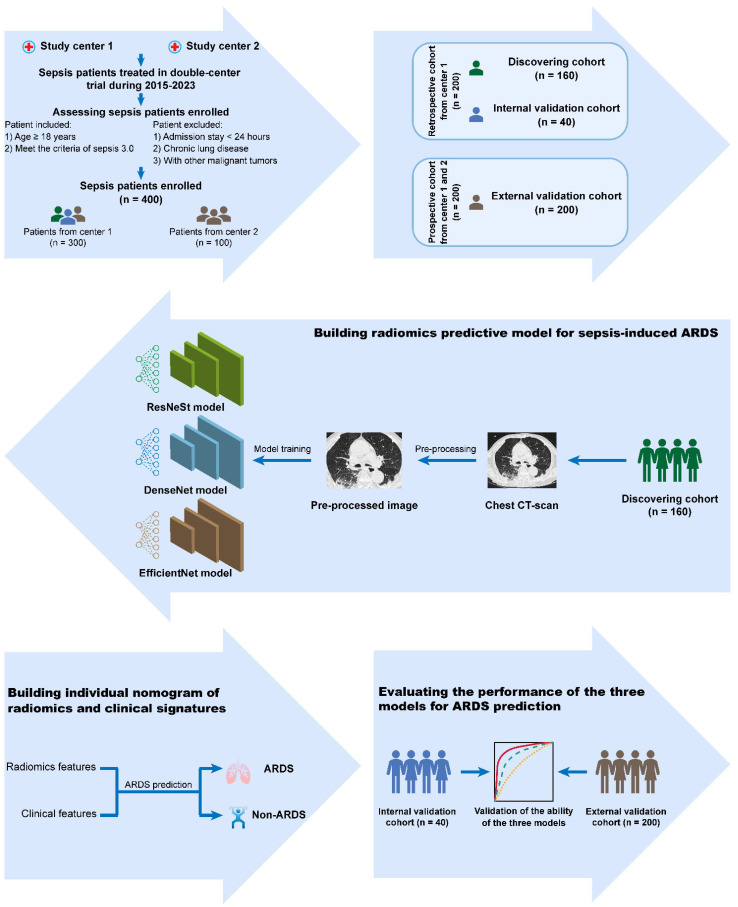
Flow chart of the study protocol.

**Figure 2 healthcare-10-02150-f002:**
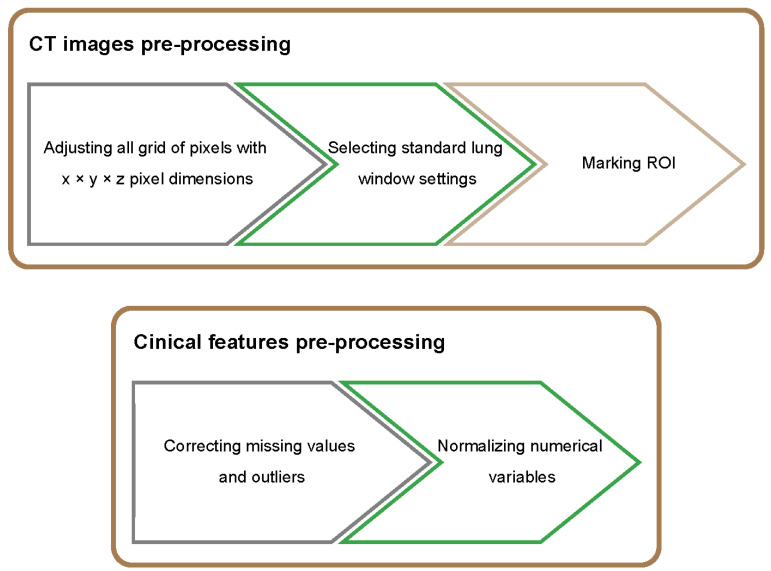
The workflow of data pre-processing.

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
