# Peer review of "Deep Learning Chest CT for Clinically Precise Prediction of Sepsis-Induced Acute Respiratory Distress Syndrome: A Protocol for an Observational Ambispective Cohort Study"

_healthcare, 2022, doi:10.3390/healthcare10112150_

Round 1

Reviewer 1 Report

The Authors provide an interesting framework of analysis to test the role of radiomics in the prediction of sepsis induced ARDS.

The protocol seems good, and I personally find it useful the design of retrospective and prospective evaluation.

I only suggest the Authors to consider also other Institutions to test the models in a real external validation analysis.

Author Response

Response to Reviewer #1

  1. I only suggest the Authors to consider also other Institutions to test the models in a real external validation analysis.

Response: We thank the editor for the questions. We agree that using an independent validation cohort from a different institution would be worthful to test our model performance. Therefore, we consider to include 100 patients from Shenshan Medical Center as an external validation cohort to test the models. As the affiliated hospital of Sun Yat-sen Memorial Hospital, Sun Yat-sen University, Shenshan Medical Center, Memorial Hospital of Sun Yat-sen University is located in Shanwei, east of the central coast of Guangdong Province. This hospital shares the same superior medical resources of Sun Yat-sen Memorial Hospital, Sun Yat-sen University, so the eligible patients included in the hospital are quite suitable for the external validation of our model. We have added detailed cohort information in the manuscript (Lines 37-52, page 2) and modified our workflow shown in Figure 1. 

Reviewer 2 Report

In this manuscript, researchers have introduced the work on deep learning on chest CT. Essentially, a protocol has been demonstrated for an observational ambispective cohort study. The technical writing has been prepared well. "2.4 Data collection" can be improved. Instead of listing, full description will be better for the journal manuscript writing. Only one figure is shown in the writing. "Revision" is suggested.

Author Response

Response to Reviewer #2

  1. "2.4 Data collection" can be improved. Instead of listing, full description will be better for the journal manuscript writing.

Response: We thank the editor for the questions. We have re-writed "2.4 Data collection" and fully described how we collected the patient’s data. (Lines 21-32, page 4).

  1. Only one figure is shown in the writing.

Response: We thank the editor for the questions. We have added one more figure as Figure 2 to present our study protocol more clearly.

Reviewer 3 Report

Thank you for giving me the opportunity to review this article. The retrospective cohort study using CT image data was first performed in this study to pick up possible risk factors for SI-ARDS. After that, the external validation test was done using other patients' data sets. This protocol seemed to be interesting. However, I think many points need to be corrected. I'm afraid I think this article does not reach the journal's acceptable level.   The followings are my comments. -This paper has no result description. -Many sentences are written in the present form. And the word "will" is frequently used. Research papers, particularly their methods and results, are regularly described in the past form.

Author Response

Response to Reviewer #3

  1. This paper has no result description.

Response: We thank the reviewer for the questions. However, since the follow-up of the relevant patients, data collection and data analysis are still continuing, we have not completed the baseline characteristics and are still unable to describe our study results and draw the eventual conclusion.

  1. Many sentences are written in the present form. And the word "will" is frequently used. Research papers, particularly their methods and results, are regularly described in the past form.

Response: We thank the reviewer for the questions. We are sorry for our incorrect writing in the present and future tense. Considering your suggestion, we have carefully revised the paper and corrected our writing mistakes.

Reviewer 4 Report

Dear authors,

I have written your manuscript with pleasure.

However I have some doubts, which have risen while assessing the document.

Most importantly taking into account that the recrutation lasted for several years, and is already completed, and the fact that that according to authors the follow-up of the included patients will be completed in April 2023. It seems unjustified to publish the protocol now. I would appreciate an attempt to publish this study protocol directly after completion of retrospective patient inclusion, but now, when the recrutation has finished  and patient observation will be completed in few months,  I would suggest to published original data when only it is available.

Additional bur slight problem is due to the BC acceptance. The authors write that  Among the 300 patients, 200 of them were retrospectively recruited from May 2015 to June 2021, and 100 patients were prospectively recruited from  June 2021 to May 2022;  This study has been approved on August 19, 2020 by the Ethics Committee of Sun Yat-sen Memorial Hospital, Sun Yat-sen University, Guang Zhou, China.- please explain

Currently even for retrospective analyses BC committee approval or written information that it is not is required is mandatory for the manuscript to be published. potential , but in the manuscript there is data on this, was the retrospective part assessed /accepted by the BC?

Author Response

Response to Reviewer #4

  1. Most importantly taking into account that the recrutation lasted for several years, and is already completed, and the fact that that according to authors the follow-up of the included patients will be completed in April 2023. It seems unjustified to publish the protocol now. I would appreciate an attempt to publish this study protocol directly after completion of retrospective patient inclusion, but now, when the recrutation has finished and patient observation will be completed in few months, I would suggest to published original data when only it is available.

Response: We thank the reviewer for the questions. We agree that publishing original data when only it is available would be helpful. However, since this research requires large workload, and the whole research process is relatively long, we believe that showing our overall study design to the public is quite meaningful, so as to acquire professional advice for the improvement of this protocol, such as the feasibility of our study design and the quality of study data. Meanwhile, the publication of this protocol can be recognized by peers, which may facilitate the communication between peers and further enhance our research. In addition, It takes some time to complete data collection and further analysis to get the final conclusion. After the above consideration, we hope to publish this protocol first.

  1. Additional bur slight problem is due to the BC acceptance. The authors write that Among the 300 patients, 200 of them were retrospectively recruited from May 2015 to June 2021, and 100 patients were prospectively recruited from June 2021 to May 2022; This study has been approved on August 19, 2020 by the Ethics Committee of Sun Yat-sen Memorial Hospital, Sun Yat-sen University, Guang Zhou, China.- please explain

Response: We thank the reviewer for the questions. We are sorry for our inappropriate inclusion time for retrospective cohort and prospective cohort. As suggested by the reviewer, we have changed the inclusion time of the retrospective cohort the prospective cohort. Among the 300 patients, 200 of them were retrospectively recruited from May 2015 to August 2020, and 100 patients were prospectively recruited from August 2020 to May 2022. (Lines 37-46, page 2)

  1. Currently even for retrospective analyses BC committee approval or written information that it is not is required is mandatory for the manuscript to be published. potential, but in the manuscript there is data on this, was the retrospective part assessed /accepted by the BC?

Response: We thank the reviewer for the questions. The retrospective part was assessed and accepted by the Ethics Committee of Sun Yat-sen Memorial Hospital, Sun Yat-sen University, Guangzhou, China. We have submitted the ethical approval certification as a supplemental document.

Round 2

Reviewer 3 Report

Thank you very much for sending me the second version. The manuscript has been revised well. I think this revised manuscript is suitable for acceptance.

Reviewer 4 Report

Dear authors,

Dear Authors,

As mentioned before I do disagree with the time of study protocol publication, when the recreation and observations are almost completed. On the other hand I do appreciate the study design, therefore I decided to leave the decision to the editorial office. I'm looking forward in seeing the original data of your research. 

Best Wishes,